Technological trends in epidemic intelligence for infectious disease surveillance: a systematic literature review

Kamarul Aryffin Hazeeqah Amny 1 hazeeqahamny123@raudah.usim.edu.my
http://orcid.org/0009-0007-6628-2644 Bin Sahbudin Murtadha Arif 2
Ali Pitchay Sakinah 1 3 sakinah.ali@usim.edu.my
Abhalim Azni Haslizan 1 3
Sahbudin Ilfita 4
1 Faculty of Science and Technology, Universiti Sains Islam Malaysia , Nilai, Negeri Sembilan , Malaysia
2 Institute of Applied Data Analytics, Universiti Brunei Darussalam , Bandar Seri Begawan , Brunei
3 CyberSecurity and Systems (CSS) Research Unit, Faculty of Science and Technology, Universiti Sains Islam Malaysia , Nilai, Negeri Sembilan , Malaysia
4 Rheumatology Research Group, Institute of Inflammation and Ageing, University of Birmingham , Birmingham , United Kingdom
Alatas Bilal
Electronic publication date: 2025 May 6
Publication date: 2025
Volume: 11
Electronic Location ID: e2874
Received 2024 Aug 15; Accepted 2025 Apr 14
Copyright: © 2025 Kamarul Aryffin et al.
Copyright year: 2025
Copyright holder: Kamarul Aryffin et al.
License: This is an open access article distributed under the terms of the Creative Commons Attribution License, which permits unrestricted use, distribution, reproduction and adaptation in any medium and for any purpose provided that it is properly attributed. For attribution, the original author(s), title, publication source (PeerJ Computer Science) and either DOI or URL of the article must be cited.
License URL: https://creativecommons.org/licenses/by/4.0/

Keywords: Epidemic intelligence, Artificial intelligence, Big data, Internet of things, Geographic information systems, COVID-19, Public health surveillance

Funding: Ministry of Higher Education (MOHE), Malaysia under the Fundamental Research Grant Scheme FRGS/1/2023/ICT03/USIM/02/3 and USIM/FRGS/FST/KPT/51723 This research was funded by the Ministry of Higher Education (MOHE) Malaysia under the Fundamental Research Grant Scheme (FRGS/1/2023/ICT03/USIM/02/3) with Research Code (USIM/FRGS/FST/KPT/51723). The funders had no role in study design, data collection and analysis, decision to publish, or preparation of the manuscript.

==============================
Background

This research focuses on improving epidemic monitoring systems by incorporating advanced technologies to enhance the surveillance of diseases more effectively than before. Considering the drawbacks associated with surveillance methods in terms of time consumption and efficiency, issues highlighted in this study includes the integration of Artificial Intelligence (AI) in early detection, decision support and predictive modeling, big data analytics in data sharing, contact tracing and countering misinformation, Internet of Things (IoT) devices in real time disease monitoring and Geographic Information Systems (GIS) for geospatial artificial intelligence (GeoAI) applications and disease mapping. The increasing intricacy and regular occurrence of disease outbreaks underscore the pressing necessity for improvements in public health monitoring systems. This research delves into the developments and their utilization in detecting and handling infectious diseases while exploring how these progressions contribute to decision making and policy development, in public healthcare.

Methodology

This review systematically analyzes how technological tools are being used in epidemic monitoring by conducting a structured search across online literature databases and applying eligibility criteria to identify relevant studies on current technological trends in public health surveillance.

Results

The research reviewed 69 articles from 2019 to 2023 focusing on emerging trends in epidemic intelligence. Most of the studies emphasized the integration of artificial intelligence with technologies like big data analytics, geographic information systems, and the Internet of Things for monitoring infectious diseases.

Conclusions

The expansion of publicly accessible information on the internet has opened a new pathway for epidemic intelligence. This study emphasizes the importance of integrating information technology tools such as AI, big data analytics, GIS, and the IoT in epidemic intelligence surveillance to effectively track infectious diseases. Combining these technologies helps public health agencies in detecting and responding to health threats.

Introduction

Intelligence can be generally defined as the process of gathering and analyzing information to gain insights. This forms the basis of epidemic intelligence which is a field that focused on identifying health risks (Bowsher, Milner & Sullivan, 2016; Hughbank & Githens, 2010). Epidemic intelligence integrates structured data analysis with real-time information to support public health actions (Yan, Chughtai & Macintyre, 2017).

In the fight against the epidemic, modern tools like big data and artificial intelligence (AI) are now widely used to address epidemics in areas like healthcare, transportation, and education (Ahmad et al., 2023; Jiao et al., 2023). The COVID-19 pandemic has impacted various industries; therefore, health organizations, including the World Health Organization (WHO), have suggested various precautions to address and control the rapid worldwide spread of COVID-19 quickly (Ahmad et al., 2023; MacIntyre et al., 2023). The Epidemic Intelligence from Open Sources (EIOS) developed by WHO combines both existing and new surveillance systems worldwide (Raina MacIntyre et al., 2023; Shausan, Nazarathy & Dyda, 2023).

The COVID-19 pandemic has demonstrated the need for technology in managing public health crises. The rapid spread of the virus emphasized the drawbacks of traditional monitoring methods, which are time-consuming for public health authorities (laboratories and hospitals) as they need to gather infectious disease data primarily through positive laboratory tests, as well as records of hospitalizations and fatalities (Ganser, Thiébaut & Buckeridge, 2022; MacIntyre et al., 2023; Shausan, Nazarathy & Dyda, 2023). The conventional approach is slow and lacks real-time capabilities, prompting the adoption of digital technologies to track disease spread, model epidemiological patterns, and aid in public health decision-making (Hii et al., 2018). Real time infectious diseases monitoring is crucial for developing both immediate and prolonged public health strategies and preventive actions (Lalmuanawma, Hussain & Chhakchhuak, 2020; Raina MacIntyre et al., 2023; Shausan, Nazarathy & Dyda, 2023).

Recent advancements in epidemic intelligence, particularly in AI and big data analytics have improved public health surveillance (Allam, 2020; Wong, Zhou & Zhang, 2019). While these technological progressions have strengthened surveillance systems, some gaps and difficulties require further exploration. This systematic literature review aims to evaluate and examine the enhancements in information technologies to enhance awareness and public health surveillance systems. By examining studies and real-life examples in various reports and cases, this review seeks to offer a thorough insight into the ways in which these technologies are employed to forecast and oversee public health risks ultimately assisting in improving the surveillance efforts, for public health.

The review focuses on four main objectives, including: 1. To identify the latest advanced technologies particularly in AI, big data analytics, Internet of Things (IoT) and Geographic Information Systems (GIS) and their applications in epidemic intelligence

2. To summarize the use of advanced technologies in identifying and managing outbreaks.

3. To evaluate the role of advanced technologies to support public health professionals in decision making

4. To study the effectiveness of advanced technologies in improving public health surveillance.

The study reviews the integration of technological trends mainly AI, big data, IoT and GIS in epidemic intelligence for infectious disease monitoring. It analyzes the latest advancements and application of these technologies in real time monitoring, outbreak detection, decision making and public health responses for infectious disease surveillance. These advanced technologies effectively address the traditional surveillance gaps such as delayed responses to infectious disease and inefficiency in decision making. Other than that, this study highlights the application of these technologies on the COVID-19 pandemic where the pandemic has utilized these technologies for contact tracing, combating misinformation and identifying hotspot locations. This article aims to guide future research and public health professionals to optimize the use of these technologies for proactive infectious disease monitoring.

Research methodology

This review ensures a detailed and transparent approach which involves identifying, screening, eligibility assessment, inclusion and exclusion of studies. This methodology is structured to gather and analyze findings on the use of advanced technologies in epidemic intelligence for infectious disease surveillance. The method begins with a structured search strategy to cover relevant literature across various databases, then the application of pre-defined eligibility criteria focusing on how advanced technological tools improve public health surveillance. This research methodology also includes multi-stage screening procedures for selection of studies. A process flow diagram will be utilized to transparently document the study selection process as shown in Fig. 1.

Figure 1 Process flowchart illustrating the selection process of including and excluding articles.

Literature search strategy

The search query involves combinations between key terms and the synonyms related to epidemic intelligence and technology using Boolean operators (“Epidemic Intelligence” OR “Epidemic Open Source” OR “Infectious Disease Monitoring” OR “Epidemic Prevention” OR “Disease Surveillance”). Additional technological aspects factor is included in the search within the Science Direct and PubMed databases as follows: (“Epidemic Intelligence” OR “Epidemic Open Source” OR “Infectious Disease Monitoring” OR “Epidemic Prevention” OR “Disease Surveillance” AND “technology”). This query is designed to review a wide range of studies while being sufficiently specific to target the use of technological tools in the context of epidemic surveillance and intelligence.

To ensure the search captures the most relevant and up-to-date research, filters will be applied where necessary to include articles published in English, from the year 2019 onwards. This time frame is chosen to reflect the period during which technological advancements have been most rapidly evolving and influencing public health surveillance strategies. The initial search will be complemented by backward reference search, which involves reviewing the reference lists of included studies to identify additional sources that could be missed through database searching.

Inclusion and exclusion criteria

The eligibility criteria for inclusion in this review are defined to ensure the selection of studies that are directly relevant to the use of technological tools in epidemic intelligence for public health surveillance. Studies will be included if they meet the following criteria: (1) they focus on the application of technology in epidemic intelligence, particularly in AI, big data analytics, mobile technology, the IoT or GIS; (2) they present original research, including case studies, observational studies, intervention studies, and analyses; and (3) they are published in peer-reviewed journals or conference proceedings. Exclusion criteria will include non-English articles, articles published before the year 2019, low quality journal, editorials, and studies not focusing on technological applications in public health surveillance.

Further eligibility assessment will be based on the study’s relevance to the systematic review’s objectives. Studies must provide empirical data or detailed analysis on the effectiveness, challenges, or outcomes of deploying technological tools in epidemic surveillance and prevention efforts. Theoretical papers, unless they provide significant insight into the application of technology in public health surveillance, will be excluded. Additionally, the key details from each study are being categorized into each section of advanced technologies such as AI, big data analytics, IoT and GIS. These sections were further subdivided into each of their applications in epidemic intelligence for infectious disease monitoring. Figure 1 provides a detailed summary of the 69 studies reviewed on the use of advanced technological tools to improve epidemic intelligence systems for monitoring infectious diseases.

Public health response with epidemic intelligence from open sources (EIOS)

This study presents the initial findings in a series of research that extends the work published in the Malaysian Journal of Science Health & Technology (MJOSHT), titled Advancing Localized Public Health Surveillance in Malaysia by Enhancing EIOS with Google COVID-19 Data Integration. EIOS was studied to further understand on how they can improve epidemic intelligence and public health monitoring. The current article sets the stage for further research and offers a deeper understanding of how technological trends can strengthen disease surveillance systems.

Epidemic intelligence is important for public health monitoring in real time. EIOS is an initiative from the WHO to improve infectious disease surveillance system for COVID-19. WHO, together with the European Commission’s Joint Research Centre (JRC) utilizes EIOS in both existing and new systems to improve public health surveillance (MacIntyre et al., 2023). It gathers and analyzes open-source data to get information on how the disease spreads and identify hotspots area.

Analysis process of EIOS frameworks and algorithms

EIOS uses its frameworks and algorithms to improve accuracy of analyzing open-source data. Figure 2 illustrates the framework used by EIOS to ensure rapid detection and containment of clusters before the outbreak escalates (Aryffin et al., 2025). Traditional surveillance does not have this capability and might cause delay in gathering and analyzing open-source data. EIOS uses NLP and text mining to process millions of multilingual news and data which are useful in identifying high risk areas and aid communication between public health professionals (MacIntyre et al., 2023). EIOS is widely used in event-based surveillance to improve preventive measures during the events held in the outbreak. EIOS also collaborates with INFORM (Index for Risk Management) to further analyze possible hotspots area to ensure timely intervention (Spagnolo et al., 2020).

Figure 2 Sequential framework of WHO-recommended public health surveillance for COVID-19 (WHO, 2022).

Data from Worldometer (WOM) that are being used in EIOS are sanitized before being displayed as shown in Algorithm 1. The process checks for missing data and handles it by replacing empty fields or “−” with “NA.” It also removes commas and plus signs to convert the data into integers while managing any encountered issue. Whenever new data is fetched automatically, it goes through the same cleaning, validation, and transformation process.

Algorithm 1 Data cleaning and validation in WOM.

1.  Start	
2.    for each column i, from 2 to n do	
3.     If data[i] = "" or data[i] = "−"	
4.       Set data[i] = NA	
5.     Endif	
6.     remove any ’,’ from data[i]	
7.     remove any ’+’ from data[i]	
8.     // convert data from other format to integer	
9.     tmp = convert data[i] to integer	
10.      // Check for any issues during conversion	
11.      If tmp has warning then	
12.       Set data[i] = original format	
            // If conversion is unsuccessful, restore data to its original format	
13.      Else	
14.       Set data[i] = tmp	
             // Convert the data in column i to an integer	
15.      Endif	
16.    Endfor	
17.  End	

The findings from the data sanitization and validation process employed in EIOS ensure data integrity and reduce errors. In conventional systems, data handling is inconsistent, which can lead to misinterpretation of information (Kaur et al., 2021). EIOS’s algorithmic approach to data cleaning ensures that any missing data is addressed systematically.

Algorithm 2 calculates the daily new COVID-19 cases in Malaysia. The algorithm is summarized by Eq. (1):

(1) ΔC(t)=C(t)−C(t−1)

Algorithm 2 Daily new cases (∆C) for COVID-19.

1.   Start	
2.     for each day i = 1 to n do	
3.      If i = 1 then	
4.        new_covid19_cases[i] = total_covid19_cases[i]	
5.      Else	
6.         new_covid19_cases[i] = total_covid19_cases[i] – total_cumulative_cases[i − 1]	
7.      Endif	
8.    Endfor	
9.  End	

In Eq. (1), ΔC(t) represents the change in the number between current day cases, C(t) and previous day cases, C(t − 1).

Algorithm 3 calculates the total COVID-19 cases in Malaysia. The algorithm is summarized by Eq. (2):

(2) C(t)=C(t−1)+ΔC(t)

Algorithm 3 Cumulative COVID-19 cases (C(t)).

1.   Start	
2.     for each day i = 1 to n do	
3.      If i = 1 then	
4.        total_covid19_cases[i] = new_covid19_cases[i]	
5.      Else	
6.        total_covid19_cases[i] =	
             total_covid19_cases[i−1] + newcovid19_cases[i]	
7.      Endif	
8.    Endfor	
9.  End	

In Eq. (2), C(t) represents current’s total cases, C(t − 1) represents previous day’s total cases, and ΔC(t) represents today’s new cases.

The findings show that EIOS’s ability to track daily new cases (Algorithm 2) and cumulative cases (Algorithm 3) dynamically improves decision-making processes. Traditional methods often rely on manual reporting, which may not reflect the real-time status of an outbreak. The algorithms used in EIOS ensure that the latest data is always available, supporting timely decision-making and intervention. This ability to utilize real-time data is in line with the technological trends in epidemic intelligence, which emphasizes quick responses and early detection in infectious disease surveillance.

Trends in epidemic intelligence

Epidemic intelligence provides quick response and early detection in infectious disease surveillance (Bhatia et al., 2021; Ganser, Thiébaut & Buckeridge, 2022). One of the notable frameworks was developed by WHO as shown in Fig. 2. The frameworks and systems are developed to ensure rapid detection of infectious diseases and to provide containment of clusters before the outbreak escalates further. Traditional public health surveillance systems rely heavily on manual reporting, laboratory studies and field investigations which may hinder the process of detecting and responding to the outbreak as it does not provide data in real time (WHO, 2022). Therefore, the major advancements of public health surveillance capabilities lie in the increase in open sources data which is being used timely to improve accuracy of detection and prediction (Agbehadji et al., 2020; Shausan, Nazarathy & Dyda, 2023).

Infectious diseases can spread through direct contact, consumption of contaminated food or water, airborne particles, and vectors which are usually spread by insects and animals (Baker et al., 2022). Nowadays, infectious disease can easily spread between countries as people do a lot of travelling, therefore it is important to have advanced epidemic intelligence systems to detect and quickly response to infectious diseases to prevent further spread of the disease quickly (Ahmad et al., 2023; Mboussou et al., 2019; Morgan & Pebody, 2022). Huang et al. (2020) highlighted a dramatic change in travel data during the COVID-19 pandemic due to the effectiveness of containment measures. Their studies show the change in transportation-related factors, such as changes in preferred modes of travel, visited places patterns, and a preference for local, zero-contact travel.

The well-known trend in epidemic intelligence is artificial intelligence. It utilizes machine learning (ML) and natural language processing (NLP) to analyze open sources data which are beyond human capabilities to identify anomalies in data patterns. The next trend is in big data analytics, which analyzes vast open sources of data and is used for outbreak predictions, resource allocation and provide a more targeted intervention (Agbehadji et al., 2020; Ahmad et al., 2023; Wong, Zhou & Zhang, 2019), while GIS helps improve spatial data analysis to identify a more precise hotspot locations and at-risk populations (Shausan, Nazarathy & Dyda, 2023; Stanko, 2023).

This systematic literature review integrates current trends on the use of technological tools in epidemic intelligence mainly on artificial intelligence, big data analytics, internet of things and GIS. Infectious diseases are becoming more complex, therefore advanced technologies are required to quickly response to the outbreak preventive measures can be deployed widely. This review aims to provide insight on how advanced technological tools can be utilized to optimize public health surveillance which is essential to guide future researchers, policy makers and the development of technology-driven surveillance systems.

Application of artificial intelligence

AI focuses on developing a system that can mimic human intelligence. The core foundation of AI is the algorithms that enables machines to do self-learning, reasoning and correction. AI has been widely used in epidemic intelligence. Knowledge-based system filter available data using expert defined terms to help in complex decision-making. NLP assists in human-computer communication as it can understand and interpret multiple human languages. ML allows machines to learn by themselves and improve their performance without the need for human interaction. ML is used to analyze patterns and provide results based on experience and available data (Babel et al., 2021; Comito & Pizzuti, 2022; Firouzi et al., 2021; Ganasegeran & Abdulrahman, 2020; MacIntyre et al., 2023; Pillai & Kumar, 2021; Shausan, Nazarathy & Dyda, 2023). It helps health professionals in making decisions and improve predictive capabilities in epidemic intelligence. Deep learning is one of the subsets in ML where it uses neural networks with multiple layers to analyze complex patterns and data. It is widely used in epidemic intelligence for image analysis, outbreak predictions and disease diagnosis (Comito & Pizzuti, 2022; Firouzi et al., 2021; Pillai & Kumar, 2021).

NLP and automatic speech recognition (ASR), key branches of AI, have seen substantial development and use across various applications for organizing and structuring knowledge. They contribute significantly to telemedicine consultations, aiding medical professionals and citizens in safe and convenient remote healthcare services (Jiao et al., 2023). Moreover, these technologies have become crucial in intelligent conference systems, facilitating seamless remote communication during events like online meetings, especially event during the COVID-19 pandemic (Jiao et al., 2023; Kasamatsu et al., 2021; Yanagawa et al., 2022). They have also contributed to the development of vaccines and drugs, analyzed, and learned from data, text, and images, conducted clinical, pathological, and genomic analysis, and facilitated intervention and management (Cao, 2022; Firouzi et al., 2021). However, each technique brings its own set of benefits and challenges as simplified in Table 1, suggesting that a multi-faceted approach may be most effective in leveraging AI for infectious disease surveillance and response.

Table 1 Summary of the study focus area.

Field (No. of studies)	Focus area	Topics	Remarks	
Artificial intelligence (18)	1. Machine learning (Babel et al., 2021; Cao, 2022; Comito & Pizzuti, 2022; Firouzi et al., 2021; Ganasegeran & Abdulrahman, 2020; Jiao et al., 2023; Karaarslan & Aydın, 2021; Kaur et al., 2021; Lalmuanawma, Hussain & Chhakchhuak, 2020; MacIntyre et al., 2023; Nguyen et al., 2021; Pillai & Kumar, 2021; Shausan, Nazarathy & Dyda, 2023; Stanko, 2023; Wong, Zhou & Zhang, 2019; Zeng, Cao & Neill, 2020)

2. Natural language processing (Cao, 2022; Ganasegeran & Abdulrahman, 2020; Jiao et al., 2023; MacIntyre et al., 2023; Pillai & Kumar, 2021; Raina MacIntyre et al., 2023; Shausan, Nazarathy & Dyda, 2023)

3. Deep learning (Cao, 2022; Chae, Kwon & Lee, 2018; Comito & Pizzuti, 2022; Ganasegeran & Abdulrahman, 2020; Jiao et al., 2023; Nguyen et al., 2021; Pillai & Kumar, 2021)

4. Image analysis (Ganasegeran & Abdulrahman, 2020; Jiao et al., 2023)

	Early detection and prediction	1. Machine learning algorithms identify patterns for early detection (Firouzi et al., 2021; Ganasegeran & Abdulrahman, 2020; Jiao et al., 2023; MacIntyre et al., 2023; Rahman et al., 2022b).

2. Early warning systems to improve outbreak prediction (Shausan, Nazarathy & Dyda, 2023).

	
Data analysis and decision support	1. Analyzing large datasets to support decision-making (Jiao et al., 2023).

2. Real-time AI analysis enables rapid public health responses (Rahman et al., 2022b; Shausan, Nazarathy & Dyda, 2023).

	
Predictive modeling for outbreak detection	Machine learning algorithms analyze data patterns to prevent outbreak (Shausan, Nazarathy & Dyda, 2023).	
Big data analytics (18)	1. Big data analytics (Batko & Ślęzak, 2022; Chae, Kwon & Lee, 2018; Jiao et al., 2023)

2. Blockchain (Ahmad et al., 2023; Alo et al., 2022; Bragazzi et al., 2020; Chattu et al., 2019; Firouzi et al., 2021; Garg et al., 2020; Hang et al., 2022; Karaarslan & Aydın, 2021; Kaur et al., 2021; Marbouh et al., 2020; Nguyen et al., 2021; Upadrista, Nazir & Tianfield, 2023; Xie et al., 2021)

3. Bayesian Change Point (BCP) analysis (Ahmad et al., 2023)

	Data sharing	Employs asymmetric cryptography to ensure the genuineness and integrity of data (Ahmad et al., 2023).	
Countering misinformation	Blockchain based platform to detect fake information (Ahmad et al., 2023).	
Contact tracing	Utilizing anonymized data collection (Ahmad et al., 2023).	
Outbreak detection	Real-time data sharing by blockchain technology (Ahmad et al., 2023).	
Internet of Things (19)	1. Mobile app (Alo et al., 2022; Firouzi et al., 2021; Kondylakis et al., 2020; Lai et al., 2019; Pillai & Kumar, 2021; Rahman et al., 2022b; Shausan, Nazarathy & Dyda, 2023; Stanko, 2023; Ye, 2020)

2. Remote monitoring devices (Dami, 2022; Firouzi et al., 2021; Gao et al., 2021; Jiao et al., 2023; Lai et al., 2019; Menni et al., 2020; Moglia et al., 2022; Rahman et al., 2022a, 2020; Saheb & Izadi, 2019; Sarker et al., 2021; Sim & Cho, 2021; Stanko, 2023; Sudre et al., 2021; Tripathy et al., 2022)

3. Radio Frequency Identifier (RFID) (Alo et al., 2022; Garg et al., 2020; Rahman et al., 2022b)

4. Near field communication (NFC) (Alo et al., 2022; Sim & Cho, 2021)

	IoT devices in disease surveillance	Enable real-time data collection and monitoring of various health parameters for disease surveillance (Alo et al., 2022; Chi Minh City et al., 2021; Rahman et al., 2022b).	
IoT implementation in rural locations	Bridge the gap in healthcare access and surveillance in remote areas (Alo et al., 2022; Shausan, Nazarathy & Dyda, 2023; Stanko, 2023).	
Geographic information system (16)	1. Geospatial artificial intelligence (GeoAI) (Kamel Boulos, Peng & Vopham, 2019; Mir et al., 2021; Peng et al., 2021; Rahman et al., 2022b; Sahana et al., 2023; Stanko, 2023)

2. Spatial patterns (Bag et al., 2020; Carballada & Balsa-Barreiro, 2021; Desjardins, Hohl & Delmelle, 2020; Franch-Pardo et al., 2020; Gangwar & Ray, 2021; Kamel Boulos, Peng & Vopham, 2019; Malone et al., 2019; Mir et al., 2021; Sarfo & Karuppannan, 2020; Sarwar et al., 2020; Scarpone et al., 2020; Xu et al., 2022; Zhou et al., 2020)

	Disease mapping	Visualizing disease spread patterns for effective surveillance and control (Carballada & Balsa-Barreiro, 2021; Gangwar & Ray, 2021).	
Potential of GeoAI in health geographics	GeoAI enhances spatial analysis capabilities for better understanding of disease dynamics and risk assessment (Mir et al., 2021).	

Data analysis and decision support

ML algorithms can estimate the size and location of infections and analyze key variables such as transmission methods, incubation periods, symptoms, and treatment resistance (Kaur et al., 2021). ML methods like convolutional neural networks, transfer learning, support vector machines, random forest, deep learning, and gradient boosting machine learning, are instrumental in predicting geospatial risks associated with outbreaks and the spread of epidemics. By analyzing multi-dimensional data that includes regional data on past outbreaks, environmental factors, travel patterns, social influences, vector distribution, and meteorological conditions such as temperature and rainfall, these models can accurately forecast the occurrence and timing of the outbreaks. This capability provides a valuable framework for early response and preparation in public health (MacIntyre et al., 2023).

ML has advanced greatly over the years and is now widely applied across various sectors. However, the success of ML depends on making optimal choices, including selecting the right algorithms, preprocessing methods, and hyperparameter settings for each dataset. Achieving adequate performance from a machine learning system requires expertise, yet even skilled professionals frequently rely on time-consuming and costly trial and error to identify the most effective methods for specific datasets (Karaarslan & Aydın, 2021).

Another successful evidence in the usage of AI in controlling epidemics is the outbreaks caused by mosquitoes such as Dengue, Chikungunya and Zika as human capabilities alone will not be enough to detect all the mosquito vectors in the affected area. During the outbreaks, AI assisted in identifying mosquito populations in the area to help improve epidemic management (Ganasegeran & Abdulrahman, 2020).

Predictive modeling for outbreak detection

AI has been helpful in handling various challenges faced by the healthcare sector such as the predictive capabilities for outbreak detection (Comito & Pizzuti, 2022). Predictive modeling uses mathematical models and formulas to calculate and predict the likelihood of the emergence and spread of epidemic (Firouzi et al., 2021; Raina MacIntyre et al., 2023). Chae, Kwon & Lee (2018) compared the long short-term memory (LSTM) model with deep neural network (DNN) models for predicting past outbreaks such as chicken pox and malaria. Additional factors that have been considered in the study include Naver search frequency, Twitter mentions, and average daily temperature and humidity. The findings show that the LSTM model performed better in disease outbreak while DNN models show consistent result whether there is an outbreak or not which makes the LSTM model is being widely used in predictive modeling (Zeng, Cao & Neill, 2020).

AI-based surveillance for early detection and prediction

AI assist health professionals by using advanced technological tools to analyze data and predict the spread of the outbreak. The COVID-19 pandemic accelerated the use of AI in epidemic surveillance. AI tools can help identify high risk locations and the duration of the outbreak such as the utilization of ML to detect outbreaks in the early stages. ML helps generate early warnings by analyzing open sources data which is beneficial in all places especially low-income countries that mostly depend on the traditional surveillance (Firouzi et al., 2021; Lalmuanawma, Hussain & Chhakchhuak, 2020; MacIntyre et al., 2023; Sim & Cho, 2021). The collection and analysis of the open sources data can quickly identify the patterns of the outbreak and take preventive measures to minimize the risk of the disease spreading (Firouzi et al., 2021; MacIntyre et al., 2023). Table 2 highlights the data analysis and predictive modeling approaches in various infectious disease surveillance systems.

Table 2 Benefits and challenges in using AI TEchniques.

Comparison of data analysis and predictive algorithms in infectious disease surveillance systems.

Epidemic intelligence system	Main features	Data analysis techniques	Predictive techniques	
Epidemic Intelligence from Open Sources (EIOS)
(MacIntyre et al., 2023)	Real time infectious disease surveillance by utilizing open sources data to detect health risks.	Utilizes NLP and text mining to analyze vast amounts of open-source data for disease trends and anomalies. NLP and keyword-based filters identify potential outbreaks.	Integrate with INFORM to provide additional predictive capabilities for crisis and risk assessment
(Spagnolo et al., 2020).	
HealthMap
(MacIntyre et al., 2023)	Real time monitoring of infectious disease outbreaks using data from various sources.	Combines ML, NLP and Fisher–Robinson Bayesian filtering to classify and process vast amounts of data. These algorithms identify, geo-locate, and display infectious disease data from open sources on an interactive map.	Utilize ML and NLP in early detection of pandemic.	
BioCaster
(Collier et al., 2008; Gu et al., 2022; Meng et al., 2022)	Collect and analyze open sources data to detect health risks.	Utilizes rule-based algorithms and ontology-based NLP to detect early signals of public health threats from online sources. Semantic web technologies support accurate categorization of health events.	Combines ML and NLP for predefined health threat categories.	
BlueDot (Allam, 2020; Allen, 2016)	Utilizes AI to track, analyze, and predict the spread of infectious diseases.	Utilizes ML, NLP and big data analytics to monitor open sources data and identifies unusual trends with anomaly detection algorithms.	Utilizes ML to simulate and project the spread of diseases geographically, incorporating factors like travel data, environmental variables, and population density.	
MetaBiota
(Allam, 2020; Tong, 2020)	Utilizes AI models to predict the spread of the outbreak.	Utilizes ML and NLP by analyzing various data sources, including epidemiological reports and environmental factors. This approach allows for probabilistic assessments of disease emergence and spread.	Employs advanced predictive models that incorporate Bayesian statistics to forecast outbreak likelihood, potential impact, and economic consequences.	

EIOS uses NLP and text-mining to process millions of multilingual news to quickly identify health risks and improve communication between health professionals (MacIntyre et al., 2023). EIOS collaborates with the Index for Risk Management (INFORM) to identify and alert locations at risk to enable timely response and intervention. EIOS is also widely used for event-based monitoring for early detection and improve preventive measures (Spagnolo et al., 2020).

HealthMap was developed in 2006. It utilized Fisher–Robinson Bayesian filtering within a Linux/Apache/MySQL/PHP framework, integrating tools like Google Maps, GoogleMapAPI, Google Translate API, and a single AJAX library for efficient operation. It uses text processing algorithms to quickly identify, classify and overlay information on interactive map. HealthMap processed approximately 80 infectious disease alerts daily to provide real time insights for health professionals. HealthMap also successfully detect unidentified pneumonic cases in Wuhan Province outbreak which later widely known as COVID-19 a day before it was officially announced by the Chinese government (MacIntyre et al., 2023).

Next, BioCaster was launched in 2008. It utilized an ontology-based text mining system which is used to detect early outbreaks and disease monitoring (Collier et al., 2008). It then relaunched in 2021 during COVID-19 with improved features such as the inclusion of NLP to process multilingual news daily and was used for epidemic surveillance and predictive modeling. BioCaster is a web-based system with a friendly user interface which visualize real time statistics and outbreak trends (Gu et al., 2022; Meng et al., 2022).

BlueDot was developed in 2009 for H1N1 influenza pandemic. It accurately predicts the movement of the virus by using air travel data. In 2014, BlueDot was utilized to predict the Ebola Outbreak in West Africa (Allen, 2016). BlueDot also predicted COVID-19 nine days prior to the official announcements of COVID-19 and successfully predicted high risk countries. It analyzes over 10,000 data using ML and over 60 languages using NLP daily to identify high risk areas and alert health professionals in real time, allowing them to respond quickly and take preventive measures before the outbreaks escalate further (Allam, 2020).

Metabiota utilized ML and NLP to predict infectious disease outbreaks. NLP is used to categorize data into clusters such as frequency, severity and duration of the outbreak. During COVID-19, it successfully predicts high risk neighboring countries such as Japan, Thailand and Hong Kong by taking human behavior and fear-driven responses into account. It also accurately foresees the spread of the COVID-19 virus 1 week before it was officially reported in these countries (Allam, 2020; Tong, 2020).

Application of big data analytics

Big data analytics refers to the process of examining large and complex datasets to identify patterns, trends, and insights that are not apparent in smaller datasets. The defining characteristics of big data are the collection of large amounts of datasets, its rapid production speed, the variety of data sources, and the needs to maintain a high-quality data.

The utilization of big data analytics in blockchain technology is recognized during the COVID-19 pandemic, especially from a healthcare emergency perspective (Ahmad et al., 2023; Bragazzi et al., 2020; Firouzi et al., 2021). Blockchain technology enhances big data analytics by providing a secure and immutable platform for data collection and sharing, ensuring the integrity and accuracy of large datasets (Bragazzi et al., 2020; Firouzi et al., 2021). By integrating blockchain, big data analytics can analyze decentralized data management, leading to improved trust and transparency in data-driven decision-making processes. Recent developments in blockchain have led to various applications ensuring data privacy while enabling COVID-19 testing remotely, digital contact tracing, and supporting remote health monitoring for outpatients (Ahmad et al., 2023; Chattu et al., 2019; Firouzi et al., 2021).

Data sharing

The existing systems manage unstructured COVID-19 data which leads to difficulties in data interoperability. Unstructured data hinders collaboration opportunities to fight against the pandemic (Kaur et al., 2021). Thus, implementing blockchain technologies can help reduce the unstructured data, as blockchain supports better information sharing and provide a structured data view, enhancing the efforts to control the spread of COVID-19 (Ahmad et al., 2023; Firouzi et al., 2021; Karaarslan & Aydın, 2021; Nguyen et al., 2021).

Blockchain technology creates a secure, constant record of transactions across a wide network across geographical settings through hashing algorithms and asymmetric cryptography (Ahmad et al., 2023). Each block in the blockchain is linked to the previous block by using cryptographic hashing, which makes the data unchangeable. There are two main types of blockchain which are permissionless and permissioned. In permissionless blockchain, anyone can participate in doing any transactions and mining data, while permissioned blockchain, is restricted to only a specific group of users chosen by the organization (Ahmad et al., 2023; Chattu et al., 2019).

Countering misinformation

Blockchain are being used widely in epidemic intelligence to combat misinformation by tracking the origin of the data and identifying unauthorized changes to the data. This ensures the reliability and accuracy of the data for public health professionals. One of the blockchain-based platforms, MiPasa supports sharing reliable data, identifying errors and misreporting by integrating data from various trusted open sources data such as the WHO and registered health organizations. It utilized data analytics and verified blockchain data to facilitate identifying hotspot locations and virus carriers while also protecting user anonymity and data privacy (Ahmad et al., 2023; Marbouh et al., 2020; Nguyen et al., 2021).

Blockchain can also verify news authenticity by registering, evaluating, and sorting news based on the credibility of the sources. Blockchain aids public health professionals in decision making which facilitates them to quickly foresee future outbreaks and implementing lockdown in affected areas (Ahmad et al., 2023; Comito & Pizzuti, 2022).

Contact tracing

A countless of mobile applications have been developed for infectious disease surveillance such as during the COVID-19 outbreak. However, the challenges arrive when using a centralized architecture as shown in Table 3, which means all the data is stored and managed in a single location. An example is Singapore’s TraceTogether, which uses Bluetooth to determine if someone has been in close contact with a COVID-19 positive person (Ahmad et al., 2023; Firouzi et al., 2021). Centralized systems raise concerns about data privacy, as authorities can access user information. They are also at risk of data tampering, fraud, or loss and are considered less reliable due to the potential for a single point of failure (Firouzi et al., 2021). Additionally, such systems may limit cooperation between different entities like healthcare, government, and law enforcement, and lack transparency, traceability, and permanence in data management during the pandemic (Ahmad et al., 2023).

Table 3 Benefits and challenges in centralized data.

Challenge	Resolution	Benefits	Limitations	
Data Privacy and Security (Ahmad et al., 2023; Firouzi et al., 2021)	1. Implement blockchain technology to ensure secure, tamper-proof, and transparent data transactions.

2. Enforce strict data encryption standards and access controls to protect sensitive information.

3. Incorporate multi-factor authentication and continuous monitoring for unauthorized access.

	1. Enhanced trust among users through immutable record-keeping, ensuring data integrity.

2. Prevention of unauthorized access, reducing potential data breaches.

3. Transparent logs enable auditability and accountability.

	1. High computational costs and energy consumption associated with blockchain implementation.

2. Complex integration into existing centralized systems, requiring significant expertise.

3. Potential legal and regulatory hurdles in adopting blockchain technology.

	
Data Interoperability (Karaarslan & Aydın, 2021; Nguyen et al., 2021)	1. Establish standardized data formats (e.g., JSON, XML) for seamless exchange between systems.

2. Develop interoperable APIs to connect disparate systems.

3. Promote the adoption of universal data exchange standards across organizations and industries.

	1. Improved collaboration and data-sharing capabilities, fostering innovation and reducing redundancy.

2. Comprehensive analyses were made possible by integrating data from diverse sources.

3. Reduced time and cost for systems integration and compatibility resolution.

	1. Achieving consensus on standardization may require extensive negotiation among diverse stakeholders.

2. High upfront costs and extended timelines for implementing standardized solutions.

3. Incompatibility with legacy systems, requiring additional adjustments.

	
Data Authenticity (Ahmad et al., 2023; Comito & Pizzuti, 2022)	1. Use advanced Natural Language Processing (NLP) techniques to detect, flag, and correct misinformation.

2. Implement blockchain for traceability and verification of data origins.

3. Employ machine learning algorithms to validate data authenticity in real-time.

	1. Enhanced ability to detect and correct false or manipulated information, improving data reliability.

2. Easier verification of trusted sources, reducing misinformation circulation.

3. Strengthened public and organizational confidence in data usage.

	1. Total elimination of misinformation remains a challenge due to the dynamic nature of its creation.

2. Risk of false positives and negatives in detection algorithms, leading to incorrect conclusions.

3. High computational and infrastructure requirements for real-time validation at scale.

	

Blockchain technology safeguards user data privacy through pseudo-anonymity (Behnaminia & Samet, 2023). Digital contact tracing can utilize regular expression matching techniques on a blockchain platform to store social interaction data securely, ensuring only authorized users can access it (Ahmad et al., 2023).

Outbreak detection

Blockchain technology facilitates immediate data exchange among healthcare providers, government bodies, and various stakeholders, enabling the quick identification of irregular patterns in disease occurrences (Hang et al., 2022; Upadrista, Nazir & Tianfield, 2023; Xie et al., 2021).

Bayesian change point (BCP) was used for outbreak detection. The BCP analysis is not initially designed for infectious disease outbreaks; however, it has been utilized to detect the beginning of influenza epidemics by identifying significant shifts in distributional parameters before and after certain points in time series data. The BCP algorithm breaks down the time series into blocks, then calculates the mean and variance, followed by estimating the probability of each break point being a change point (Ahmad et al., 2023).

Application of the Internet of Thing

The IoT is a network of interconnected devices through embedded sensors and communications technology which is used to collect. transmit and analyze data. Some examples of the devices are smartphones, wearable health monitors such as smart watch, and environmental sensors to ensure real-time data collection and analysis.

The IoT is being widely used in collection of open-source data to make further analysis such as predictive models for infectious disease outbreak (Perwej et al., 2019). Open source data collected through wearable sensors, mobile applications, crowdsourcing, social media and web search analytics has improved conventional way to real-time infectious disease surveillance (Rahman et al., 2022b; Shausan, Nazarathy & Dyda, 2023; Yanagawa et al., 2022). The data collected from IoT has been used various aspects including formulating effective public health policies and strategies for infectious disease surveillance (Firouzi et al., 2021; Stanko, 2023). Recent research has found that IoT also has been widely used in other technologies such as social distancing monitoring, managing patient quarantines, tracking the virus, remote health monitoring, sanitizing infected areas, and offering contactless healthcare treatment (Alo et al., 2022; Dami, 2022; Firouzi et al., 2021; Stanko, 2023).

IoT devices in disease surveillance

Remote monitoring devices like wearable devices, IoT sensors, and mobile apps are valuable tools for healthcare providers to remotely track a patient’s health status (Sim & Cho, 2021). These devices can gather data on vital signs, respiratory rate, and oxygen levels, aiding in the early detection of health deterioration. Table 4 presents a detailed overview of the devices used for data collection and their applications in epidemic intelligence. The availability of this data, along with advanced intelligent algorithms, has the potential to advance medical practices focusing on disease prediction, prevention, and treatment (Firouzi et al., 2021; Rahman et al., 2022b; Stanko, 2023). The market has seen a rise in wearable devices for personal health and fitness which cause healthcare professionals to explore their use in continuous monitoring, management, and accessing patient physiological information within remote health monitoring systems (Stanko, 2023). mHealth encompasses various medical and public health practices supported by mobile devices like smartphones, patient monitoring tools, and wireless gadgets (Chi Minh City et al., 2021).

Table 4 Data collected from IoT devices and its limitations.

IoT devices	Data collected	Application in public health	Benefits	Limitations	
Wearables fitness tracker (Firouzi et al., 2021)	Heart rate, activity levels, sleep patterns	1. Monitoring public health.

2. Early detection of potential COVID-19 symptoms.

	1. Enables continuous and non-invasive health monitoring.

2. Encourages proactive user engagement in personal health.

	1. Data accuracy may vary depending on device quality and calibration.

2. Raises concerns about user privacy and data security.

	
Mobile phones
(Alo et al., 2022; Chi Minh City et al., 2021; Firouzi et al., 2021)	GPS location data, health app data.	1. Contact tracing to monitor disease spread.

2. Dissemination of public health information.

3. Self-reporting of symptoms.

4. Providing remote health advice.

	1. Offers wide accessibility for data collection.

2. Potential for large-scale public health analysis.

3. Familiarity with mobile devices ensures higher adoption rates.

	1. Heavy reliance on user engagement and active participation.

2. Privacy concerns due to location and health data tracking.

3. Self-reported information may lack accuracy.

	
Smart thermometers (Ahmad et al., 2023)	Body temperatures	1. Early detection of fever and potential infection hotspots.

2. Data integration with public health systems for tracking trends.

	1. Enables real-time symptom tracking.

2. Facilitates data sharing with healthcare professionals for immediate action.

	1. Limited to measuring fever and cannot detect asymptomatic cases.

2. Requires widespread adoption for meaningful data insights.

	
Robotics (Firouzi et al., 2021; Jiao et al., 2023; Pillai & Kumar, 2021)	Temperature data, patient health data, environmental data.	1. Automated disinfection to reduce infection risks.

2. Telepresence in healthcare for remote consultations and patient monitoring.

	1. Reduces the need for human contact, minimizing risks of transmission.

2. Facilitates scalable healthcare delivery through automation.

	1. High deployment costs, limiting widespread adoption.

2. Potential technical issues, such as malfunctions or compatibility challenges.

	

Advanced tracking devices like AI-based apps in bracelets and rings can effectively trace infected individuals, aiding in outbreak detection and control measures for diseases like COVID-19 and dengue (Rahman et al., 2022b). Various tracking applications and GPS systems are used to identify patients and implement efficient control measures. Additionally, technologies such as Radio Frequency Identification (RFID), Near-Field Communication System (NFC) and AI significantly enhance disease and vector surveillance, monitor environmental and social risks related to COVID-19 and dengue, and (Alo et al., 2022; Chi Minh City et al., 2021; Rahman et al., 2022b; Sim & Cho, 2021).

Garg et al. (2020) developed a model using passive RFID technology to anonymously track the movement and interaction of objects until someone tests positive for an infectious disease like COVID-19. This system doesn’t require mobile phones; instead, individuals and animals will wear RFID tags. Their innovative approach also incorporates blockchain technology to store the data securely, ensuring privacy through decentralized control. The RFID readers are placed in buildings or vehicles to collect proximity information, which is then recorded on a blockchain-based smart contract.

Wearables

Before and during the pandemic, Shausan, Nazarathy & Dyda (2023) mentioned that a few wearables such as the TempTraq skin patch, Fitbit smartwatches, and Oura Rings have been utilized for health monitoring and early detection of conditions such as COVID-19. These devices, which track temperature, heart rate, and other physiological data, have helped healthcare workers and research studies to monitor and analyze health trends. The integration of wearable device data with self-reported symptoms and other information has enabled the development of algorithms to detect and track the progression of COVID-19 (Shausan, Nazarathy & Dyda, 2023; Sim & Cho, 2021). EPIWATCH employs two AI systems; one uses NLP for named entity recognition to identify and extract details like disease, date, and location from reports. The second AI system helps in sorting outbreak reports by relevance, filtering out non-pertinent articles, to streamline the analysis and presentation of data to users and analysts (Firouzi et al., 2021; MacIntyre et al., 2023; Raina MacIntyre et al., 2023). Proximity and vision sensors are incorporated into wearables, robots, and smart devices to ensure adherence to social distancing measures. These sensor-based systems detect proximity and alert users with a warning alarm if social distancing rules are breached (Alo et al., 2022; Firouzi et al., 2021).

Mobile phone

Mobile applications help combat infectious disease by providing information on symptoms, treatment and preventive measures to their user (Firouzi et al., 2021; Kondylakis et al., 2020; Ye, 2020). It allows individuals to monitor their own health status, receive reminders for medication and vaccination and alert health professionals about potential outbreaks which allow them to take immediate action to areas that have limited access to medical care (Alo et al., 2022; Dami, 2022; Firouzi et al., 2021; Stanko, 2023). Improvement in mHealth technology and the widespread use of mobile phones have eased the monitoring of individual’s health behaviors. Individuals with long-term illnesses can now also stay at home instead of staying in the hospitals (Saheb & Izadi, 2019). These advancements also allow for more reliable applications that collects data from various sources for further research and practice in travel medicine (Lai et al., 2019).

Smart thermometer

In the initial stages of COVID-19 pandemic, governments used traditional thermometers for temperature checks at specific sites, but this method had limitations in consistently identifying infected individuals. Advancements led to the exploration of smart thermometers and wearable devices for continuous health monitoring (Tripathy et al., 2022). These devices, which track physiological changes to signal potential illness, had been previously used in the U.S. for flu surveillance and were adapted to detect COVID-19 by analyzing data trends and physiological markers. For example, data from smart thermometers helped predict COVID-19 hotspots, while wearables like smartwatches and fitness trackers were employed to detect early signs of the virus, utilizing metrics like respiratory and heart rate changes (Shausan, Nazarathy & Dyda, 2023).

Robotics

The integration of 5G communication technology with various technologies such as drones and aids in monitoring legal compliance within the transport system. It also supports thermal screening of individuals through thermal cameras, facilitating the instant sharing of date and time-specific information (Moglia et al., 2022; Rahman et al., 2022b). Given the rapid spread of viruses, manual contact tracing is often insufficient. However, the speed and efficiency offered by 5G technology can significantly improve control measures (Pillai & Kumar, 2021).

AI-based imaging and robotic applications have become essential tools for diagnosis, treatment, and daily life support during the pandemic (Firouzi et al., 2021; Jiao et al., 2023). Imaging techniques in AI play a vital role in detecting and managing COVID-19, with contributions like masked face recognition and fever detection aiding in tracing cases and contacts (Firouzi et al., 2021; Jiao et al., 2023). The surge in video conferencing due to homeschooling and telework has led to increased use of face detection and portrait segmentation, enhancing convenience in these activities. Additionally, robotic technologies incorporating face detection, voice recognition, and sensors have been widely utilized for tasks such as examination, healthcare, monitoring, disinfection, cleaning, delivery, and logistics, reducing human contact and enhancing daily life support. Hence, the development of intelligent robots is a crucial direction for the future (Firouzi et al., 2021; Gao et al., 2021; Jiao et al., 2023; Sarker et al., 2021).

Contact tracing

IoT devices are widely used in epidemic intelligence as a contact tracing tool. It is used to track the places visited by infected individuals and identify close contacts based on that. Close contact is being guided by health professionals to protect themselves and to prevent them from further spreading the virus. The utilization of contact tracing systems has also reached rural areas to track individual movements accurately. Contact tracing systems in urban areas utilize technologies like GPS, Bluetooth, and Wi-Fi networks for authorities to quickly identify potential exposures and implement necessary preventive measures (Ahmad et al., 2023; Alo et al., 2022; Firouzi et al., 2021; Menni et al., 2020; Pillai & Kumar, 2021). Other than that, NFC, a wireless communication protocol operates over a short range of up to 4 cm at speeds around 424 kbps, was also considered as an emerging technology in contact tracing (Alo et al., 2022). Although contact tracing can be challenging due to high mobility of individuals, it is important to identify infected individuals who should be quarantined and to determine containment zones for managing disease transmission (Pillai & Kumar, 2021; Sudre et al., 2021).

Application of geographic information systems

GIS is being used to capture, store, analyze, and visualize spatial and geographic data. By analyzing this data, GIS helps identify relationships between geographic variables and anomalies in infectious disease patterns.

GIS application and geospatial analysis are crucial for mapping the spread of the virus, contact tracing, testing, and vaccine distribution, where it includes the geographic characteristics of each region (Carballada & Balsa-Barreiro, 2021; Gangwar & Ray, 2021). GIS systems facilitate dynamic mapping by integrating temporal and spatial information (Franch-Pardo et al., 2020; Gangwar & Ray, 2021). GIS helps to track and analyze infectious disease cases to predict the spread and to take preventive measures quickly (Zhou et al., 2020). GIS offers various methods to understand how the virus spreads across different areas, which helps public health officials, decision-makers, and communities make informed responses and plans (Desjardins, Hohl & Delmelle, 2020; Malone et al., 2019). Notable utilization of GIS was in 2009 during swine-origin influenza A-H1N1 and identifying hotspots location in MERS-CoV in camel. GIS was also used to predict the movement of avian influenza in wild bird populations (Carballada & Balsa-Barreiro, 2021). Table 5 presents the applications of GIS in epidemic intelligence.

Table 5 GIS application in epidemic management and limitations.

GIS application	Function	Epidemic management	Benefits	Challenges	
Disease mapping (Carballada & Balsa-Barreiro, 2021; Malone et al., 2019)	Visualization of disease spread over geographic areas.	1. Helps identify spatial patterns and disease hotspots.

2. Assists in resource allocation and developing intervention strategies.

	1. Offers a clear visual representation of complex data, aiding decision-makers.

2. Facilitates targeted public health interventions.

	1. Accuracy depends on the quality and timeliness of data.

2. Privacy concerns when handling sensitive location-based health data.

	
Contact tracing (Gangwar & Ray, 2021)	Tracking interactions between individuals to identify potential transmission chains.	1. Vital for early detection of cases and containment measures.

2. Prevents widespread transmission by isolating infected individuals.

	1. Enables rapid response to emerging outbreaks.

2. Assists in monitoring the effectiveness of public health interventions.

	1. Relies heavily on widespread participation and accurate data collection.

2. Concerns about privacy and misuse of sensitive information.

	
Mobility analysis (Mir et al., 2021)	Analyzing movement patterns to understand and predict disease spread.	1. Informs policies for lockdowns and social distancing measures.

2. Identifies critical areas for targeted interventions.

	1. Helps predict future disease hotspots, enabling proactive planning.

2. Aids in the effective implementation of public health measures.

	1. Challenges in collecting accurate movement data without infringing on privacy.

2. Requires access to advanced analytical tools and expertise.

	

Disease mapping

GIS is important in visually displaying the factors that correlate with the disease occurrence (Sarwar et al., 2020). It analyzes spatial autocorrelation, a concept that helps understand how proximity impacts the likelihood of interaction and similarity among neighboring factors to understand the spatial relationship in the spatial epidemiology. GIS observes patterns between time and space to identify how infectious diseases are linked between the socioeconomic factors of different groups and the effectiveness of various prevention and control strategies. The advancements in geospatial analysis together with GIS tools enables better handling and analyzing vast amount of data (Bag et al., 2020; Sarfo & Karuppannan, 2020; Sarwar et al., 2020; Scarpone et al., 2020).

During COVID-19, geospatial technologies gained attention through web-mapping applications and dashboards as they provide visual graphics on the outbreak’s information which are important to inform the public, raise social awareness, and manage uncertainties in pandemic (Gangwar & Ray, 2021; Xu et al., 2022). Notable dashboards like HealthMap utilized online data for real-time surveillance to enhanced spatial resolution in specific areas while WHO offered near-real-time official data, updated approximately every 15 min. Real-time mapping was essential for risk assessment and to support decision-making. GIS systems not only used for disease mapping, but also as visual platforms that display real time data and track individuals at risk (Carballada & Balsa-Barreiro, 2021).

Web apps and platforms for contact tracing help analyze the spatial patterns and spread of the virus over time. Most of the countries developed their own system, such as China’s Close Contact Detector (Carballada & Balsa-Barreiro, 2021), Singapore’s Trace Together (Carballada & Balsa-Barreiro, 2021; Pillai & Kumar, 2021), and Switzerland’s SwissCoVID (Alo et al., 2022; Carballada & Balsa-Barreiro, 2021), highlighting the significance of geospatial analysis during the pandemic (Carballada & Balsa-Barreiro, 2021). This period urged advancements in geospatial analysis, with improved computational capabilities of GIS tools enabling better handling of large datasets and enhanced analytical precision (Carballada & Balsa-Barreiro, 2021).

GIS systems are more than just mapping tools; they serve as centralized platforms for near-real-time data management, aiding in tracking both reported COVID-19 cases and individuals at risk (Carballada & Balsa-Barreiro, 2021; Franch-Pardo et al., 2020; Gangwar & Ray, 2021; Malone et al., 2019).

Potential of geospatial artificial intelligence in health geographics

Geospatial artificial intelligence (GeoAI) is a combination of AI and GIS which is being used widely in spatial health analysis. It helps understand the disease spread by using data analysis techniques to track possible viruses that may cause infectious disease using location analytics and patient travel data to find effective measures to mitigate outbreak (Mir et al., 2021). GeoAI is also being utilized to monitor and forecast disease hotspots as it understands how disease spread geographically (Kamel Boulos, Peng & Vopham, 2019; Peng et al., 2021). It analyzes location data to identify factors at specific places that might affect health outcomes across different social groups. GeoAI also combined with IoT devices to integrate personal analytics by collecting data from devices such as mobile phones and wearables sensors. It integrates metadata and satellite imagery to enable geo-database in analysis ready format (Mir et al., 2021).

Data mining and machine learning algorithms like Python and Anaconda, together with geographic knowledge, facilitate accurate prediction of health and environmental threats (Sahana et al., 2023). Geostatistical and algorithmic methods are applied to identify risks from environmental and spatial factors, assessing their impact on infectious disease routes (Mir et al., 2021).

Research limitation

This study aims to provide valuable insights on the integration of AI, big data, IoT and GIS in infectious disease monitoring. However, this study only focuses on the reporting of technological practices for infectious disease surveillance. In the medical field, the integration of technological advancements is considered crucial factors for a better diagnostic analysis and vaccine development. This study limit in comprehensive understanding of how both technological and medical advancements complement each other in addressing epidemic challenges.

Next, the limitation of this study is the heavy emphasis on the COVID-19 pandemic cases as this study focuses more on articles from the last 5 years. This may reduce the insights on other infectious diseases with contrasting characteristics and transmission dynamics. Examples of underexplored infectious diseases include influenza and those of vector-borne diseases such as dengue and malaria. Therefore, this study does not covers comprehensive conclusions for non-pandemic infectious diseases.

Additionally, practical implementation of these technologies in real-word settings require more literature and reporting. Most of these technologies involve large costs for the implementation and infrastructure. They also require comprehensive training for their workers especially in limited-resources environments. Furthermore, ethical and privacy concerns related to the use of advanced technologies in epidemic intelligence are minimally analyzed. For example, the integration of IoT devices and big data analytics often involves the collection of sensitive individual data, which may raise concerns about data security and misuse. Addressing these limitations in future research could provide a more comprehensive understanding of how advanced technologies aid in infectious disease surveillance.

Conclusion and future work

The integration of AI appeared to be the main trend in epidemic intelligence for monitoring infectious diseases. It allows analyzing large volumes of data for early detection of outbreaks and prediction of their spread. AI algorithms can sift through different open-data sources to identify potential health threats. This capability is important for timely decision-making and implementing preventive measures. Nevertheless, most of the gathered data is unstructured data with gaps and inaccuracies that demand expertise for analysis purposes.

AI powered systems heavily depend on open-source data which can lead to biases embedded in datasets mainly represent younger individuals who are more technology savvy compared to elderly (Shausan, Nazarathy & Dyda, 2023). To tackle this challenge effectively, the development of standardized and validated methods for processing open-source data, alongside the integration of advanced data-cleaning algorithms and ethical AI frameworks are required. These measures are critical to improving inclusiveness and dependability as the overall efficiency of AI powered systems utilized for epidemic monitoring and analysis. A study by Ahmed et al. (2021) on big data and IoT has proved a neural network-based model that has been used for prediction has achieved 99% accuracy. This study shows the potential of these technologies to enable quick decision making and effective pandemic management.

Next, the integration of big data analytics into epidemic intelligence to boost data interoperability is important for more efficient data exchange and collaboration among different health organizations and researchers in responding to infectious disease outbreaks. Nevertheless, the process of converting unstructured to structured data presents challenges, including maintaining data quality and managing the sheer volume of data. To address these issues, additional studies in blockchain technology are required to establish data structures that can simplify data merging and enhance compatibility among systems. Blockchain has the potential to tackle issues surrounding data privacy and trustworthiness. This could lead to an enhanced framework for epidemic intelligence and foster smoother cooperation among stakeholders.

However, there is a limit on these technologies on how widely they can be used in predicting pandemic outcomes. For example, Yan et al. (2020) developed a machine learning model to predict the risk of death based on data from 29 COVID-19 patients at Tongji Hospital in Wuhan China. Jiang et al. (2020) also developed an AI model with 80% accuracy to predict which patients might have acute respiratory distress syndrome (ARDS) based on data from 53 patients across two hospitals. These small sample sizes raise concerns on whether the findings can be used for larger populations.

Besides, the integration of AI with IoT for contact tracing significantly contributes to epidemic intelligence. Wearable devices and smartphones equipped with the ability to monitor the movement and interaction of individuals which helps to identify close contacts to ensure health authorities can take quick action to potentially at-risk individuals and infected individuals to prevent further spread of the disease. The integration of IoT in contact tracing raises concerns about privacy and data security. For instance, location records and health data could be at risk of security breaches or misuse by individuals. To address these issues, future research should explore robust privacy-preserving techniques, such as differential privacy, data anonymization, and secure multi-party computation, to safeguard user data. Blockchain technology could be used for ensuring tamper proof and decentralized sharing of data.

Other than that, GIS is also being widely used in epidemic intelligence where it visualizes and analyzes the spatial distribution of infectious diseases, thereby enhancing outbreak detection and response. Disease mapping helps identify hotspots and anomalies patterns which are crucial for identifying containment clusters and to allocate resources quickly and accurately. This spatial analysis can also reveal environmental and social factors contributing to the spread of infections. Visual representation of data through GIS makes complex information more accessible and comprehensible to the public. Future advancements should prioritize the development of systems that can seamlessly integrate real time data and enhance the interoperability with other technologies. These efforts are essential to fully discover the potential of GIS in boosting the ability to enhance the impact of public health professionals.

Therefore, the integration of AI, big data analytics, IoT, and GIS in epidemic intelligence shift towards a more dynamic and effective infectious disease monitoring. These technologies collectively enhance the ability to detect, analyze, and respond to epidemics in real-time. However, their effectiveness underlies challenges such as data privacy, ethical considerations, and the need for interdisciplinary collaboration. Moving forward, the focus should be on refining these technologies to ensure they are efficient, transparent, and ethically sound, thus maximizing their potential to safeguard public health against future epidemics.

This study of integrating advanced technologies like AI, big data analytics, GIS, and IoT in epidemic intelligence is becoming increasingly essential due to the escalating complexity and frequency of infectious disease outbreaks. Traditional surveillance methods are time-consuming and often inadequate when dealing with infectious diseases, this emphasizes the need for effective epidemic intelligence surveillance tools.

EIOS utilizes AI and big data analytics such as open-source data and advanced algorithms for infectious disease surveillance. EIOS has effectively identify the spread of virus and hotspot’s location of COVID-19 for early public health intervention. The integration of technological trends within EIOS has improved preventive measures and containment strategies for infectious disease surveillance.

This study is primarily intended for academic researchers and public health professionals who are engaged in the fields of infectious disease surveillance, public health, and epidemic intelligence. The conclusion of this study is meant to guide future research, in the advancement of better technological solutions in infectious diseases surveillance.

The COVID-19 pandemic created significant uncertainty socially, health-wise, and economically, prompting public health professionals to make decisions by sharing detailed and sensitive data with researchers. This timely, location-specific data sharing enabled complex, multi-scale analyses, leading to an improved understanding of the virus’s spatial dynamics. This research highlights the importance of utilizing these advanced technologies to combat the spread of infectious diseases in urban areas.

Additional Information and Declarations

Competing Interests

The authors declare that they have no competing interests.

Author Contributions

Hazeeqah Amny Kamarul Aryffin conceived and designed the experiments, performed the experiments, analyzed the data, prepared figures and/or tables, and approved the final draft.

Murtadha Arif Bin Sahbudin conceived and designed the experiments, authored or reviewed drafts of the article, and approved the final draft.

Sakinah Ali Pitchay conceived and designed the experiments, authored or reviewed drafts of the article, and approved the final draft.

Azni Haslizan Abhalim conceived and designed the experiments, authored or reviewed drafts of the article, and approved the final draft.

Ilfita Sahbudin conceived and designed the experiments, authored or reviewed drafts of the article, and approved the final draft.

Data Availability

The following information was supplied regarding data availability:

This is a literature review.

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
