# Peer review of "Technological trends in epidemic intelligence for infectious disease surveillance: a systematic literature review"

_PeerJ Computer Science, doi:10.7717/peerj-cs.2874_

## Round 0.1 · original submission · Major Revisions

· Academic Editor

Major Revisions

Dear Authors,

Thank you for submitting your literature review article. Reviewers have now commented on your study. Your article has not been recommended for publication in its current form. However, we encourage you to address the concerns and criticisms of the reviewers and to resubmit your article once you have updated it accordingly. When submitting the revision, please provide a clearly defined research question for this literature review paper. Furthermore, the Abstract should be attractive and contain motivation. Adding a comprehensive discussion for synthesis of findings, implications, future research, and limitations will be better.

Best wishes,

Reviewer 1 ·

Basic reporting

no comment , see attached fild

Experimental design

no comment

Validity of the findings

no comment

Annotated reviews are not available for download in order to protect the identity of reviewers who chose to remain anonymous.

·

Basic reporting

This paper addresses the growing need for enhancing epidemic intelligence systems to improve the surveillance and management of infectious diseases. Traditional surveillance methods are often inefficient, and this study reviews how advanced technologies, such as Artificial Intelligence (AI), Big Data Analytics, Geographic Information Systems (GIS), and the Internet of Things (IoT), can be integrated to optimize epidemic intelligence. The study examines 67 articles published from 2019, highlighting the increasing use of AI and other technological tools for better monitoring and control of infectious disease outbreaks.

Broad and Cross-Disciplinary Interest & Scope: The review paper is of broad and cross-disciplinary interest, particularly within the fields of public health, epidemiology, data science, and technology. it fits well within the scope of journals focusing on public health, data science, and interdisciplinary applications of technology in healthcare.

Recent Reviews in the Field: The field of epidemic intelligence, especially in the context of using AI, Big Data, GIS, and IoT, has likely been reviewed recently, particularly in light of the COVID-19 pandemic. However, the authors have justified this review by focusing on technological integration and the combined use of these tools for public health surveillance. This perspective is valuable because it emphasizes the synergies between different technological innovations and how their convergence can improve infectious disease monitoring. The paper could be accessible to both technology experts and public health professionals, making it suitable for cross-disciplinary readership.

Introduction & Audience: The introduction effectively introduces the subject, laying out the growing complexity of infectious disease outbreaks and the need for advancements in public health surveillance systems. It clearly motivates the audience by outlining the limitations of traditional methods and the potential of emerging technologies. While the target audience seems to be professionals in both public health and technological fields, the paper could better clarify whether it is aimed at policy-makers, researchers, or practitioners in health surveillance. Expanding on who stands to benefit the most from this integration (e.g., low- and middle-income countries, health organizations) might improve audience targeting.

Experimental design

Survey Methodology: The survey methodology appears consistent with a comprehensive and unbiased approach, as the authors mention using a structured search across online literature databases and applying predefined eligibility criteria to identify relevant studies.

However, the keywords used are limited. For example, 'Epidemic' was included, but 'COVID-19' was not. Missing interchangeable keywords may lead to a significant reduction in the number of relevant papers from the AI community. For instance, the top AI conference KDD held a Health Day in 2020 (https://www.kdd.org/kdd2020/health-day/index.html) focusing on COVID-19, where all the publications are related but not included in this review.

Organization of the Review: The review appears to be logically organized, discussing the integration of various technologies and their impact on epidemic intelligence. However, the topics are very broad, and the discussion for each is brief. This results in limited insights and shallow conclusions regarding these works.

Validity of the findings

I didn't review the discussion for all the papers, but based on the ones I read, I found the claim to be fair.

The trend of using AI for infectious disease surveillance is noted. However, there is a lack of in-depth discussion on how AI techniques can be deployed in the real world, and if not, what further efforts are needed.

Reviewer 3 ·

Basic reporting

The paper examines the challenge faced during enhancing epidemic intelligence systems with advanced technologies for more effective infectious disease surveillance. Concern of the paper is important, but the presentation of the paper should be enhanced to improve the readability of the paper. Please consider these comments carefully:
1) Please re-consider the “epidemic intelligence” in the title. Intelligent approaches for epidemic control is the major concern of the paper. The title should reflect the key contributions of the paper. Please improve the language of the paper and avoid using non-academic vocabularies such as many. There are some typo mistakes.
2) The paper analyses the artificial intelligence, big data, geographic information systems and IoT. Thus, the paper should be divided into these sections and each section should clearly state the recently applied methods, achievements and further requirements for the future. Key gaps and methods should be clear.
3) Abstract of the paper should state the importance, gaps, method and the key results of the paper. Introduction is long and raw. It should analyse the recent and related works with their advantages and disadvantages. The gaps should be clear and the claimed contributions should be well justified.
4) Uncertainties in the pandemic casualties and their roles should be discussed. The uncertainties can be internal or external, parametric or non-parametric.
5) Please note that recently advanced uncertain artificial intelligence-based long-term prediction and pandemic control approaches are developed which should be addressed. One can see these recent and related ones: Priority and age specific vaccination algorithm for the pandemic diseases: A comprehensive parametric prediction model. Linear and non-linear dynamics of the epidemics: System identification based parametric prediction models for the pandemic outbreaks.
6) Please also provide brief results of the research.
7) Tables and charts could help understtanding of the paper.
8) More focus sub-sections should be provided.
9) Predictive modeling for outbreak detection section should be enhanced by considering the suggestions above and further.

Experimental design

Please see the comments above.

Validity of the findings

Please see the comments above.

Additional comments

none

---

## Round 0.2 · Minor Revisions

· Academic Editor

Minor Revisions

Dear Authors,

Thank you for submitting your revised article. Feedback from the reviewers is now available. It is not recommended that your article be published in its current format. However, we strongly recommend that you address the issues raised by Reviewer 2 and Reviewer 3, and resubmit your paper after making the necessary changes.

Best wishes,

Reviewer 1 ·

Basic reporting

no comment

Experimental design

no comment

Validity of the findings

no comment

Additional comments

I have reviewed the revised version of the manuscript and carefully evaluated the changes made by the authors in response to the initial review.

I am pleased to confirm that the authors have addressed all the comments and suggestions provided during the initial review process. The revisions are satisfactory, and the manuscript now meets the necessary standards for publication

I therefore recommend the manuscript for acceptance.

·

Basic reporting

This submission is a revised version of a previous one. The revisions, as suggested by peer review, have significantly improved the quality of the paper. The scope is clearer, the structure is better, and the included references are broader.

Experimental design

The study design is reasonable.

Validity of the findings

The impact of this paper is fair

Additional comments

I appreciate the authors' effort in revising the paper. Although the quality has significantly improved, I still think the scope of the paper is too broad, while the discussion lacks depth. A limitations section is needed.

Reviewer 3 ·

Basic reporting

Unfortunately, the paper has not been revised as requested. Majority of the comments either not understood properly or they are ignored. Please enhance the critical review of the recent and related works. Please highlight the related background theory, key outcomes, advantages and disadvantages. Figures should be more informative and equations should be formed properly with enriched insights. References are not quite relevant and recent. Literature should be reviewed from a wider perspective. Please address the previous concerns carefully in addition to these ones.

Experimental design

Please see the comments above.

Validity of the findings

Please see the comments above.

Additional comments

Please see the comments above.

---

## Round 0.3 · Minor Revisions

· Academic Editor

Minor Revisions

Dear Authors,

It is imperative that the concerns of Reviewer 3 are addressed with clarity, taking into consideration the comments made previously.

Best wishes,

·

Basic reporting

I have no further comments for this revision.

Experimental design

The design is good.

Validity of the findings

the findings are insightful

Reviewer 3 ·

Basic reporting

I do not think that the paper has been revised and improved at all. The paper is raw, it does not serve the purpose of the research.

Experimental design

Lacks of insights.

Validity of the findings

Not clearly analysed and presented.

---

## Round 0.4 · accepted · Accept

· Academic Editor

Accept

Dear Authors,

The reviewer who requested a revision has not yet submitted his/her review within the time limit. I have personally evaluated the revision and am satisfied with the current version. The paper has undergone significant improvement and is now deemed ready for publication.

Best wishes,